# Predictive Markers of Crohn’s Disease in Small Bowel Capsule Endoscopy: A Retrospective Study of Small Bowel Capsule Endoscopy

**DOI:** 10.3390/jcm11154635

**Published:** 2022-08-08

**Authors:** Juho Mattila, Teppo Stenholm, Eliisa Löyttyniemi, Jukka Koffert

**Affiliations:** Department of Gastroenterology, Turku University Hospital, P.O. Box 52, 20521 Turku, Finland

**Keywords:** SBCE, small bowel capsule endoscopy, Crohn′s disease, fecal calprotectin, FC, CECDAI

## Abstract

To distinguish between functional gastrointestinal disorders like irritable bowel syndrome (IBS) and mild small bowel Crohn′s disease (CD) can be a burden. The diagnosis of CD often requires small bowel capsule endoscopy (SBCE). The main goal of this research was to find predictive markers to rule out clinically significant small bowel CD without SBCE. A retrospective study of 374 patients who underwent SBCE for suspected small bowel CD in Turku University Hospital in 2012–2020 was conducted. We gathered the patient′s laboratory, imaging and endoscopic findings at the time of SBCE. SBCE findings were graded along CECDAI (Capsule Endoscopy Crohn’s Disease Activity Index)-scoring system. Fecal calprotectin (FC), serum albumin and ESR were significantly different with patients diagnosed with CD and those with not. Hb and CRP had no significant differences between the two groups. Sensitivity, specificity, PPV and NPV for FC < 50 ug/g were 96.4%, 19.6%, 34.6% and 92.5% and for CECDAI (cut-off value 3) 98.2%, 90.3%, 81.1% and 99.1%, respectively. A CECDAI-score of 3 would be a reasonable cut-off value for small bowel CD. Small bowel CD is possible with FC < 100 ug/g. Our results suggest a follow-up with FC before SBCE for patients with no endoscopic ileitis, negative imaging results and FC < 50 ug/g before SBCE.

## 1. Introduction

Crohn′s disease (CD) is a chronic inflammatory bowel disease (IBD) that can affect the whole gastrointestinal tract. The diagnosis of CD is based on symptoms, laboratory findings, imaging findings, endoscopy findings and histopathology of biopsies. Typically, CD affects the colon and distal ileum, but 43–60% of patients have CD only in small bowel [1]. Typical symptoms of CD in small bowel are diarrhea, weight loss, abdominal pain and bloating. In the absence of weight loss, the differential diagnosis between CD and functional disorders like irritable bowel syndrome (IBS) based on symptoms only is difficult. The course of these two diseases is completely different; IBS does not lead to severe complications, but CD can lead to severe complications due to progressive inflammation, and immunosuppressive medication is often needed to stop the process. The treatment of IBS and other functional bowel disorders focuses on relieving symptoms in the absence of inflammation. Thus, the diagnosis of CD should be actively considered.

While the diagnosis of CD can be simple in the upper gastrointestinal tract or in the colon as they can be easily assessed in endoscopy, the large area of the small bowel is difficult to access in conventional endoscopy. Magnetic resonance enterography (MRE) can be used to detect severe lesions like transmural inflammation, stenosis, fistulae or extraintestinal abscesses [2,3]. MRE is not sensitive enough to detect luminal lesions in the small bowel [4,5,6,7]. Although it may lack specificity, small bowel capsule endoscopy (SBCE) is a noninvasive and sensitive procedure for finding even smaller superficial lesions in the small bowel. Indeed, over 10% of healthy subjects demonstrate mucosal breaks and erosions in their small bowel. Thus, the SBCE findings of mucosal lesions of the small bowel are not alone sufficient to establish a diagnosis of CD [8] In general, the diagnosis of small bowel CD should be established based on SBCE and MRE findings together with symptoms and laboratory findings [9,10].

Along with the lack of specificity, SBCE in general has a 2.1% risk of capsule retention. In patients with suspected CD the rate is 1.2% [11]. The main contraindication for SBCE is suspected or known small bowel stenosis [12]. SBCE procedure itself is time consuming for both the patient and the clinician, so there is a need to find better estimates of pretest CD probability.

The biomarkers conventionally used for suspected CD are decreased hemoglobin (Hb) due to iron deficiency, elevated C-reactive protein (CRP), erythrocyte sedimentation rate (ESR) and lowered plasma albumin (Alb). The most important fecal biomarker of CD is fecal calprotectin (FC), although a clear cut-off-value to rule out CD remains unclear [13,14,15,16].

Patients with small bowel stenosis suspected by symptoms, endoscopic findings or imaging present more clear cases of CD and are thus not eligible for SBCE due to capsule retention risk. These patients present complicated CD and are easier to diagnose than patients with milder luminal small bowel CD. In this study, we aimed to create threshold levels for fecal calprotectin to rule out small bowel CD in uncomplicated patients and find typical characteristics of CD patients in endoscopy and small bowel imaging. Our aim was also to find predictive markers for patient selection for SBCE to avoid unnecessary procedures.

## 2. Materials and Methods

### 2.1. Patient Selection

In this study we retrospectively collected patient data for all the patients in the SBCE archive of Turku University Hospital Department of Gastroenterology, A total of 766 SBCE procedures were performed between 5 January 2012 and 22 October 2020. A number of 374 patients selected for this study had SBCE performed for suspicion of CD. Patients who had SBCE performed for previously known small bowel CD, obscure gastrointestinal bleeding, suspicion of malignancy or refractory coeliac disease were ruled out. Only the findings in first SBCE were noted if patients had more than one SBCE performed during the follow-up period (Figure 1).

### 2.2. SBCE Procedure

SBCE was performed with Pillcam^TM^ SB2 in 2012–2013 and Pillcam^TM^ SB3 from 2013 onwards (Medtronic, Minneapolis, MN, USA). Polyethene Glycol (PEG) preparation was used prior the procedure from 2015 onwards. The SBCE findings for all the patients in this study group were reviewed by a single experienced clinician and graded according to the CECDAI-system (Capsule Endoscopy Crohn′s Disease Activity Index). The CECDAI-system is a validated scoring system of mucosal injury in Crohn′s disease of the small bowel, which is easily reproduceable and independent between interpreters [17,18,19,20].

### 2.3. Data Collection

The laboratory values (hemoglobin, C-reactive protein, serum albumin, erythrocyte sedimentation rate and fecal calprotectin) prior to SBCE procedure were collected, and if not available, the closest values after SBCE were used. The patient′s smoking status and prior use of non-steroidal anti-inflammatory drugs (NSAID) was retrieved from the electronic patient database. The previous endoscopy findings were collected, including both the macroscopic view on terminal ileum (ileitis or normal) and the histopathology of ileal biopsies (normal, nonspecific ileitis or granulomas). If patients had undergone previous imaging, the modality (MRE or computed tomography (CT) of small bowel) and the findings were gathered and classified as normal, unspecific inflammation and active inflammation. The findings in SBCE were collected and classified as CD of small bowel and normal or other. Data on capsule retention was collected. The symptoms and indications for SBCE as described in patient history by referring physician were retrieved.

The patient’s medical history after SBCE was also retrieved. The medications used were collected and classified in four categories, glucocorticoids for less than three months, glucocorticoids for more than three months, immunomodulatory drugs (thiopurines or methotrexate) and biologics (adalimumab, infliximab, ustekinumab or vedolizumab). The starting date was noted and the time from SBCE procedure date to the starting of medication was calculated. Crohn’s surgery or endoscopic dilatation incidence was noted as well. The operation dates were noted and the time from SBCE procedure date to operation was calculated. We also collected patients who were later diagnosed with CD in small bowel despite normal initial findings in SBCE.

Study data were collected from Turku University Hospital electronic patient database and managed using REDCap electronic data capture tools hosted at Turku University.

### 2.4. Statistical Methods

Continuous variables are summarized using mean and standard deviation (SD), range and 95% confidence intervals. Categorical variables are presented with counts and percentages.

The association between the CD group and non-CD groups and NSAID, biopsies, imaging findings and smoking status was tested with Fisher′s exact test. Laboratory levels were compared between the CD group and non-CD groups with Wilcoxon’s rank sum test. Additionally, the comparison between males and females of age, CD, CECDAI, immunosuppressive medication, biologics, dilatation, Crohn’s surgery and FC were performed with Wilcoxon rank sum test. The same method was used when comparing CECDAI levels between ileitis status (yes/no), or whether inflammation was active or suspected, or the biopsy results (normal, unspecific inflammation, granulomas).

Sensitivity, specificity, positive predictive value and negative predictive value was calculated for FC (using cut-offs 50, 75 and 100), ileitis in endoscopy, abnormal imaging finding, suspected inflammation in CT, active inflammation in CT, abnormal findings in CT, suspected inflammation in MRE, active inflammation in MRE and abnormal findings in MRE. Furthermore, the area under ROC curve (AUC) was calculated using trapezoidal rule and tested against 0.5 value.

*p*-values less than 0.05 (two-tailed) were considered as statistically significant. The data analysis for this paper was generated using SAS software, Version 9.4 of the SAS System for Windows (SAS Institute Inc., Cary, NC, USA).

## 3. Results

### 3.1. Patient Characteristics

Of the 374 patients included in this study, 110 (29.4%) were diagnosed with CD in small bowel in SBCE (CD group) and 264 (70.6%) not diagnosed with CD (non-CD group). The mean age of the patients who went through SBCE was 40.8 years and there was no significant difference in age between CD and non-CD group (Table 1). The age distribution in our population seemed to follow normal distribution. In total 216 (57.8%) of patients included in the study were female. In the CD group 56 (50.9%) of the patients were female and in the non-CD group 160 (42.8%) were female (Table 1). Females were diagnosed with small bowel CD at a younger mean age, 37.8 years, than males, 43.3 years.

There was no significant difference between the two groups in smoking or in NSAID-use (Table 1). Fifteen of the patients in the CD group (13.6%) had previously been diagnosed with CD in any part of the gastrointestinal tract other than small bowel and twenty-one (7.95%) of patients in the non-CD group had CD in any other part of the gastrointestinal tract.

The most common indications for SBCE in our study population were suspected CD (59.9%), diarrhea (56.7%) and abdominal pain (41.7%). High FC was mentioned as an indication for SBCE in 40.6% of the patients. Bloody stools and anemia were less frequent indications for SBCE. We found no significant differences in indications for SBCE between the CD and non-CD group.

### 3.2. Prior Lower Endoscopy

Prior to SBCE, colonoscopy was performed on 97.9% of all the patients in the study and in 93.4% of the patient’s ileum was intubated. The time between colonoscopy and SBCE was 140 d median, 239 d mean (29–785 d 95% Cl (confidence level), STD (standard deviation) = 337 d). Ileitis was observed in 70.2% of the patients in the CD group and in 24.4% of the patients in the non-CD group. In the CD group, ileal biopsies revealed unspecific inflammation in 49.0% of the patients, granulomas only in 2.94% of patients and biopsies were normal in 48.0% of the patients. In the non-CD group, ileal biopsies revealed unspecific inflammation in 20.1% of the patients, granulomas in 0.46% of the patients and biopsies were normal in 79.5% of the patients. Biopsies were significantly different between the two groups, *p* < 0.0001 (Table 2). The sensitivity of macroscopic ileitis was relatively low, 66.3%, but specificity was 75.6%, positive predictive value (PPV) was 54.3% and negative predictive value (NPV) was 83.7%.

### 3.3. Capsule Retention

In our study population the capsule retention was observed in six patients (1.6%). In the CD group the rate of capsule retention was 2.7% (three patients) and in the non-CD group 1.1% (three patients) respectively.

### 3.4. Prior Imaging

75.1% of the patients in our study population had small bowel imaging performed before SBCE. 59.9% had undergone MRE and 15.2% had undergone abdominal CT. The time between imaging and SBCE was 84.5 d median and 154 d mean (22–490 d 95% Cl, STD = 195 d). In the CD group, active inflammation was found in 24.7% of the patients, inflammation was suspected in 29.2% of the patients and the imaging findings were normal in 46.1% of the patients. In the non-CD group, active inflammation was found in 6.8% of the patients, inflammation was suspected in 20.8% of the patients and the imaging findings were completely normal in 72.4% of the patients (Table 2). The imaging findings were significantly different between the two groups, *p* < 0.0001. In this study, small bowel imaging (MRE and small bowel CT combined) had a sensitivity of only 65.2% and a specificity of 72.4% (PPV 52.2%, NPV 81.8%).

### 3.5. Laboratory Findings

There was no significant difference in Hb and Hb seemed to follow normal distribution in both CD and non-CD group. Median CRP was slightly higher in the CD group than in the non-CD group, but there was no significant difference. Median ESR was significantly higher in the CD group (9 mm/h vs. 6 mm/h) than in non-CD group. The median Alb levels were significantly lower in the CD group than in the non-CD group. The time between blood samples and SBCE were 49 d median and 67 d mean (0–191 d 95% Cl, STD = 81 d).

Median FC was significantly higher in the CD group than in the non-CD group (354 ug/g vs. 132 ug/g), *p* < 0.001 (Table 3). In patients with CD, FC seems to have a slight positive correlation for higher CECDAI-score (Spearman’s correlation = 0.23, *p* 0.014) Higher FC seems to be related to the need for biologics but not for immunosuppressive medication. The median delay between fecal calprotectin and SBCE was 94 d and mean 121 d (4–317 d 95% Cl, STD = 126 d).

In our study population, there were thirteen (11.8%) patients with diagnosed small bowel CD with FC < 100 ug/g. Ten (9.1%) patients had FC < 75 ug/g and four (3.6%) patients had FC < 50 ug/g. Eleven of these thirteen (84.6%) patients had later or earlier had FC > 100 ug/g. Sensitivity and specificity were calculated for different cut-off values of FC. FC < 100 had sensitivity of 89.0% and specificity 45.4% (PPV 41.6%, NPV 90.4%). FC < 50 had a sensitivity of 96.4% and a specificity 19.7% (PPV 34.6%, NPV 92.4%), respectively. As suspected, elevated FC has low specificity and positive predicted value. ROC-curve for FC < 50 showed an AUC of only 0.69 (*p* < 0.0001) (Figure 2).

### 3.6. Multivariate Analysis

To rule out small bowel CD without SBCE we tested several combinations of laboratory values, imaging and endoscopy findings. For simplicity, we created a combined variable model for patients with no ileitis in ileocolonoscopy and normal findings in small bowel imaging. These two were combined to FC-values. In our population patients with no ileitis and normal findings in imaging and FC < 50 ug/g, there were 2 (7.4%) patients out of 27 who were diagnosed with CD. If FC was 75 ug/g or less, in this combined model, 4 out of 60 patients (6.7%) were diagnosed with small bowel CD and if FC was 100 ug/g or less, 6 out 101 (5.9%) patients were diagnosed with small bowel CD.

### 3.7. CECDAI-Score

The CECDAI-score was significantly higher in the CD group (mean 11.6 (10.11–13.03 95%CI, median 9) than in the non-CD group (mean 0.49 (0.34–0.64 95%CI), median 0), *p* < 0.001. 98.2% of patients in the CD group had a CECDAI of 3 or more, whereas 90.5% of patients in the non-CD group had a CECDAI of less than 3. PPV and NPV for CECDAI cut-off 3 were 81.1% and 99.1%. The ROC-Curve showed AUC 0.99 for CECDAI (*p* < 0.0001) (Figure 3). The sum of sensitivity and specificity calculated from ROC-curve were the highest between CECDAI-score of 2.5 and 3.5.

Ileitis in ileocolonoscopy seems to correlate with CECDAI-score 3 or higher. Males also had higher CECDAI-score in SBCE than females, 13.3 and 9.95, but there was no statistical difference. 131 patients in the entire population had ileitis in ileocolonoscopy and the patients with ileitis had higher a CECDAI-score 6.82 mean vs. 1.89 mean patients with no ileitis (*p* < 0.0001). Patients with normal findings in imaging had mean CECDAI 3.42, patients with suspected inflammation had higher mean CECDAI (3.85) and patients with active inflammation in imaging had the highest mean CECDAI (7.68). Biopsies also seemed to have a correlation with CECDAI-score, the mean CECDAI-score for patients with normal biopsies was 2.70 vs. 6.65 with unspecific inflammation vs. 10.75 with granulomas.

### 3.8. Follow-Up

The mean follow-up time for the whole population was 1370 d (STD = 937 d), median 1350 d. Only two patients (0,8%) in our population were later diagnosed with CD in the small bowel with later SBCE after initially having normal findings in SBCE. Both patients had significantly elevated FC at the time of the initial SBCE. In CD group, four patients went through Crohn’s surgery during the follow up and only three underwent endoscopic dilatation. Men were more likely started on immunosuppressive medication (84%) than females (69%) but there was no statistical difference (*p* = 0.11). Men also had more biologic medication than females, 40.8% vs. 25.9%, but there was also no significant statistical difference (*p* = 0.14). There were no significant differences in rates of endoscopic dilatation and Crohn’s surgery between males and females.

Of the thirteen patients in CD group that had FC < 100 ug/g prior to SBCE, 12 (92%) needed glucocorticoid therapy for more than 3 months. Eight (62%) of these patients needed long term immunosuppressive medication and three (23%) patients were started on biologics. Only one patient in this subgroup went through Crohn’s surgery due to perianal disease and none of these patients required endoscopic dilatation. Three patients in CD group had FC < 50 ug/g, one of these had later FC > 50 ug/g. Two of these patients have always had FC < 50 ug/g, but they both presented a clinically mild course of CD, as they did not need immunosuppressive or biologic medication.

## 4. Discussion

This study further underlines our experience that the differential diagnosis, especially between luminal small bowel CD and functional disorders like IBS, is very difficult without SBCE. There were no significant differences in the indications and thus symptoms of the patients between the CD and non-CD groups. Colonoscopy findings and small bowel imaging also had relatively low sensitivity and a negative predictive value considering small bowel CD, which is in line with previous studies comparing SBCE and MRE [6,7].

The only laboratory tests with a significant difference between the CD and non-CD groups were FC, Alb and ESR (Table 3). Common laboratory tests used in CD-diagnostics like Hb and CRP had no significant differences between the CD and non-CD groups although Egea-Valenzuela and his colleagues found a significant difference in CRP [21].

FC is a simple and noninvasive diagnostic test, FC less than 50 ug/g had sensitivity of 96.4% for small bowel CD but specificity is poor (19.7%). The ROC-curve for FC had a weak AUC (0.69) due to high median FC in the non-CD group, but the patients in non-CD group were also suspected of having CD and were not healthy controls. This result is in line with previous studies with similar settings but with a smaller population [20,21,22]. A significant proportion of patients with FC < 100 ug/g needed immunosuppressive and biologic therapy. This also suggests a lower cut-off than the 100 ug/g suggested in a study by Koulaouzidis et al. in 2012 [20]. Based on our results a cut-off value of 50 ug/g could be used to rule out active CD in the small bowel and could be used as a threshold for patient selection for SBCE at least if the patient’s symptoms suggest functional disorders rather than CD.

SBCE is time consuming for both the patient and clinician, increases costs and is not entirely risk free, so the correct patient selection is crucial. In order to find the right patients eligible for SBCE the clinician should consider the patients symptoms, laboratory, colonoscopy and imaging findings.

Females were diagnosed with small bowel CD at a younger age and their CECDAI-score seemed higher. Although incidence of CD is generally higher in females after childhood [23], this leads to the thinking that males may have sought medical help later than females and may have suffered from symptoms for a longer time.

No clear cut-off for CECDAI-score has been established, although values ranging from 3.8–5.8 have been suggested [20,24]. In our study, the highest sensitivity and specificity on the ROC-curve was between CECDAI 2.5 and 3.5. For simplicity, we suggest that a CECDAI-score of 3 would be a reasonable cut-off value for small bowel CD. That corresponds to for example a few aphthae with diameter of less than 0.5 cm in a single segment of the small bowel, suggesting small bowel CD [18].

Patients with known strictures or symptoms suggesting small bowel obstruction did not undergo SBCE due to the risk of capsule retention; thus, most of the patients in CD group did not have severe penetrating CD, which is a lot easier to diagnose in general. The observed capsule retention rate of 1.6% for all the patients and 2.7% for patients in the CD group correlates reasonably well with the results in recent systematic reviews [11,25].

The low rates of Crohn′s surgery and endoscopic dilatation during follow-up can also be explained by patient selection.

The limitations of this study mainly depend on the study being retrospective and a single center study. The patients included in the study all had clinical suspicion of small bowel CD and thus patients in the non-CD group do not represent a population of healthy controls. Most of the patients in the Southwest Finland region with CD in small bowel are treated in the Turku University Hospital’s outpatient clinic and so the electric medical records were available, but it is possible that some of the patients may have moved to other districts or had their CD treated in some other institution, so the medication data may not be correct or was not available for all patients. This is though unlikely due to our healthcare system, where the treatment of small bowel CD is centered on Turku University Hospital.

The follow-up time differs between patients; patients who had SBCE performed later had a much shorter follow up-time. Due to the retrospective design, the time between colonoscopy, imaging and SBCE varied a lot. For laboratory results, the fecal calprotectin assay has changed during the follow-up period. In the beginning, the range was 50–2000 ug/g but later changed to 20–6000 ug/g. The CRP range also changed; in the beginning the minimum value was 10 mg/L and later it decreased to 1 mg/L. Hb and Alb also have different normal ranges for patients of different age and sex, which has not been taken into account in our study model.

The strengths of this study rely on quite a large population. To our knowledge, this is the only study that includes the follow-up of the patient population with data on medication and operations combined with initial laboratory, endoscopic and imaging findings.

To conclude, we found out that clinically relevant small bowel CD is possible even with FC < 100 ug/g. We suggest a follow-up with FC before SBCE for patients with no endoscopic ileitis, negative imaging results and FC < 50 ug/g. The CECDAI scoring system can be used to assess SBCE findings for suspected small bowel CD and a score of 3 would be a reasonable cut-off value for small bowel CD.

## Figures and Tables

**Figure 1 jcm-11-04635-f001:**
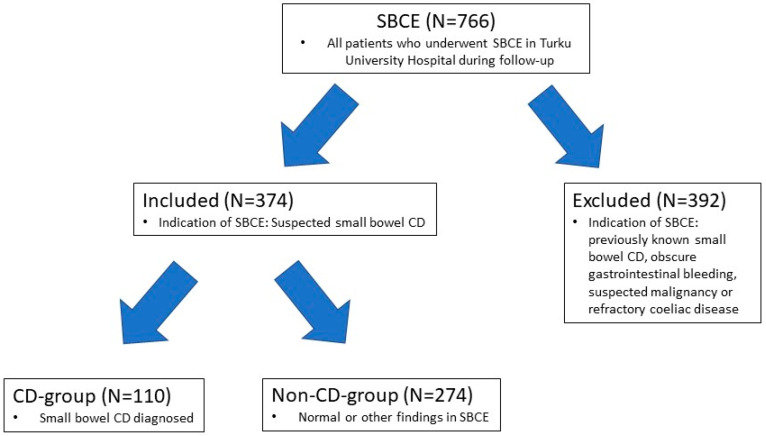
Flow-chart of study design. A retrospective study was conducted for all patients who underwent SBCE (small bowel capsule endoscopy) in the Turku University Hospital between 5 January 2012 and 22 October 2020. Patients whose indication for SBCE was suspected small bowel CD (Crohn′s disease) were included and patients with other indications for SBCE were excluded. Included patients were divided in two groups based on whether small bowel CD was diagnosed or not. Patient history with laboratory, endoscopic and imaging findings was collected for included patients and analyzed.

**Figure 2 jcm-11-04635-f002:**
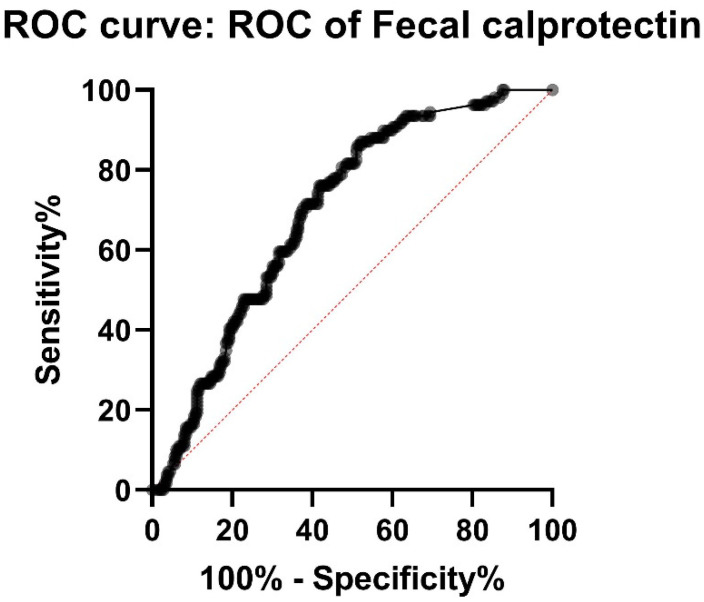
ROC (receiver operating characteristic curve) for FC (fecal calprotectin) shows AUC (area under curve) of only 0.69, meaning relatively low sensitivity and specificity of FC in the diagnostics of small bowel CD (Crohn′s disease).

**Figure 3 jcm-11-04635-f003:**
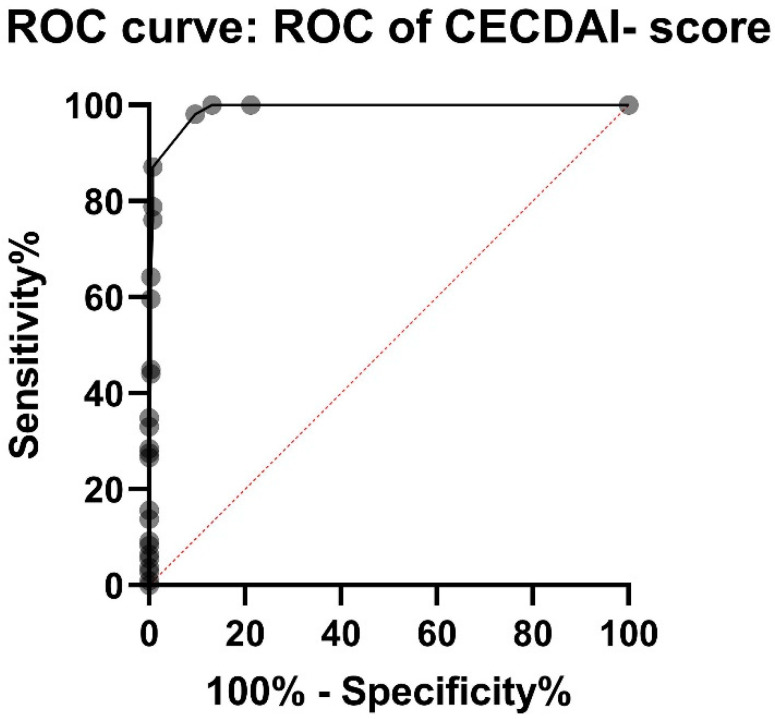
ROC (receiver operating characteristic) curve for CECDAI (Capsule Endoscopy Crohn’s Disease Activity Index) shows AUC (area under curve) of 0.99. Sensitivity and specificity were the highest between CECDAI-score of 2.5 and 3.5, suggesting a cut-off-value of 3 for small bowel Crohn’s disease.

**Table 1 jcm-11-04635-t001:** Demographics of the patients included in the study.

	All Patients (N = 374)	CD Group (N = 110)	Non-CD Group (N = 264)	*p*-Value
Characteristic	No. (%)	No. (%)	No. (%)	
Age				
Mean (Y)	40.78	40.47	40.91	0.69
95% CI (Y)	39.07–42.49	37.15–43.79	38.91–42.92	
STD (Y)	16.83			
Median [Q1:Q3] (Y)	37.28 [27.06:53.41]	36.12 [26.37:52.45]	38.05 [27.41:53.86]	
Age (min-max) (Y)	3.92–87.34	10.85–79.47	3.92–87.34	
Gender				
Male	158/374 (42.2)	54/110 (49.1)	104/264 (39.4)	0.086
Female	216/374 (57.8)	56/110 (50.1)	160/264 (60.1)	0.086
NSAID				
NSAID use	81/313 (22.5)	22/109 (20.18)	59/251 (23.51)	0.58
Smoking status				
Smokers	71/313 (22.68)	21/102 (20.59)	50/211 (23.7)	0.39
Ex-smokers	38/313 (12.14)	16/102 (15.69)	22/211 (10.43)	
Never smoked	204/313 (65.18)	65/102 (63.73)	139/211 (65.88)	

Y, year; Cl, confidence limit; STD, standard deviation; Q1, lower quartile; Q3, upper quartile; NSAID, nonsteroidal anti-inflammatory drugs.

**Table 2 jcm-11-04635-t002:** Endoscopic, imaging and biopsy findings and capsule retention rate by group.

	All Patients (N = 374)	CD Group (N = 110)	Non-CD Group (N = 264)
Finding	No. (%)	No. (%)	No. (%)
Colonoscopy			
Data available	366/374 (97.86)	109/110 (99.09)	257/264 (97.34)
Ileum intubated	342/366 (93.4)	104/109 (95.41)	238/257 (92.61)
Ileitis	131/342 (38.3)	73/104 (70.91)	58/238 (24.37)
Normal pathology in ileal biopsy	223/321 (69.5)	49/102 (48.04)	174/219 (79.45)
Nonspecific inflammation in ileal biopsy	94/321 (29.3)	50/102 (49.02)	44/219 (20.09)
Granulomas in ileal biopsy	4/321 (1.2)	3/102 (2.94)	1/219 (0.46)
Imaging			
Imaging of small bowel performed	281/374 (75.1)	89/110 (80.91)	192/264 (72.73)
Magnetic resonance enterography	224/374 (59.9)	76/110 (69.09)	148/264 (56.06)
Computed tomography	57/374 (15.2)	13/110 (11.82)	44/264 (16.67)
Normal findings	180/281 (64.1)	41/89 (46.07)	139/192 (72.4)
Suspected inflammation	66/281 (23.5)	26/89 (29.21)	40/192 (20.83)
Active inflammation	35/281 (12.5)	22/89 (24.72)	13/192 (6.77)
Capsule retention			
Capsule retention rate	6/374 (1.6)	3/110 (2.73)	3/264 (1.14)

**Table 3 jcm-11-04635-t003:** Laboratory findings by group.

	All Patients (N = 374)	CD Group (N = 110)	Non-CD Group (N = 264)	*p*-Value
Laboratory Finding	No. (%)	No. (%)	No. (%)	
C-reactive protein				
Data available	263/374 (70.6)	77/110 (70.0)	186/264 (70.5)	
Mean (mg/L) [95%Cl]	8.11 [5.9–10.3]	10.08 [5.4–14.7]	7.29 [4.9–9.7]	0.24
STD	17.84			
Median [Q1:Q3]	2 [1:8]	3 [1:9]	2 [1:8]	
[Min–max]	[1–131]	[1–131]	[1–122]	
Hemoglobin				
Data available	365/374 (97.6)	109/110 (99.0)	256/264 (97.0)	
Mean (g/L) [95%Cl]	138.3 [136.8–139.9]	138.32 [135.5–141.1]	138.3 [136.4–140.2]	0.59
STD	15.18			
Median [Q1:Q3]	139 [130:149]	138 [127:149]	139 [131:148]	
[Min–max]	[87–183]	[103–183]	[87–177]	
Erythrocyte sedimentation rate				
Data available	121/374 (32.4)	43/110 (39.1)	78/264 (29.5)	
Mean (mm/h) [95%Cl]	11.23 [9.14–13.33]	13.67 [10.05–17.30]	9.88 [7.31–12.45]	0.022
STD	11.63			
Median [Q1:Q3]	7 [2.00:14]	9 [5:20]	6 [2:12]	
[Min–max]	[1–54]	[2–48]	[1–54]	
Serum albumin				
Data available	162/374 (43.3)	72/110 (65.5)	140/264 (53.0)	
Mean (g/L) [95%Cl]	38.21 [37.68–38.74]	37.58 [36.62–38.54]	38.35 [37.89–39.17]	0.025
STD	3.93			
Median [Q1:Q3]	38.75 [36.35:40.40]	38.1 [35.6:40.1]	39.4 [37.2:40.6]	
[Min–max]	[12–46.8]	[18.7–46.8]	[15–45.3]	
Fecal calprotectin				
Data available	358/374 (95.7)	109 (99.1)	249/264 (94.3)	
Mean (ug/g) [95%Cl]	506.34 [415.38–597.30]	625.85 [496.61:755.30]	453.98 [336.15–571.81]	<0.001
STD	875.1			
Median [Q1:Q3]	207 [57:527]	354 [195:802]	132 [50:413]	
[Min–max]	[20–6000]	[22–3165]	[20–6000]	

CI, confidence interval; STD, standard deviation; Q1, lower quartile; Q3, upper quartile.

## Data Availability

Study data were collected from Turku University Hospital electronic patient database and managed using REDCap electronic data capture tools hosted at Turku University.

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
