# Peer review of "Predictive Markers of Crohn’s Disease in Small Bowel Capsule Endoscopy: A Retrospective Study of Small Bowel Capsule Endoscopy"

_jcm, 2022, doi:10.3390/jcm11154635_

Round 1
Reviewer 1 Report
Thank you for the quality of your work on a large cohort
You allow to specify non-consensual data in the literature on the thresholds of FC, ileitis, and CECDAI score
You can specify in the introduction that making the diagnosis between CD and IBS is also important because the treatments are different, beyond the risk of evolution towards complicated CD
I have no other suggestions for publication
Author Response
Reviewer 1. Thank you for the quality of your work on a large cohort
You allow to specify non-consensual data in the literature on the thresholds of FC, ileitis, and CECDAI score
You can specify in the introduction that making the diagnosis between CD and IBS is also important because the treatments are different, beyond the risk of evolution towards complicated CD
I have no other suggestions for publication
Response 1: We thank the reviewer for these opinions. We have now rewritten this part of the introduction and emphasized the importance of differential diagnosis between CD and IBS due to very different treatments. (Rows 34-38) All the revisions suggested by both reviewers were made using “Track changes” in Microsoft Word.
Reviewer 2 Report
This retrospective study intend to find predictive markers to rule out clinically significant small bowel CD without proceeding to SBCE. And they found that some useful predictive markers such as FC, small bowel imaging, and CECDAI can rule out small bowel CD. But there are still some problems in this manuscript.
1. The presentation of the methods and results sections lacks hierarchy. It is recommended to add subheadings to each section.
2. I suggest comparison with previously published relevant literature such as “Fecal calprotectin and C-reactive protein are associated with positive findings in capsule endoscopy in suspected small bowel Crohn's disease” and “A prospective study of fecal calprotectin and lactoferrin as predictors of small bowel Crohn's disease in patients undergoing capsule endoscopy”.
3. Clearly write the conclusion of the abstract and text. And the content of the abstract should correspond to the content of the text.
4. The table should be in the form of a three-line table. And the content of Table 3 is not very clear.
5. Discussion section is too scattered and needs to be more focused.
Author Response
Reviewer 2. This retrospective study intend to find predictive markers to rule out clinically significant small bowel CD without proceeding to SBCE. And they found that some useful predictive markers such as FC, small bowel imaging, and CECDAI can rule out small bowel CD. But there are still some problems in this manuscript.
- The presentation of the methods and results sections lacks hierarchy. It is recommended to add subheadings to each section.
Response 1. We thank the reviewer for this valuable opinion. We have now largely restructured both the methods and results section. Methods section is now divided into more logical sections with descriptive subheadings for easier reading. Results section was rewritten and some parts were moved from the discussion section to the results. All the revisions suggested by both reviewers were made using “Track changes” in Microsoft Word.
- I suggest comparison with previously published relevant literature such as “Fecal calprotectin and C-reactive protein are associated with positive findings in capsule endoscopy in suspected small bowel Crohn's disease” and “A prospective study of fecal calprotectin and lactoferrin as predictors of small bowel Crohn's disease in patients undergoing capsule endoscopy”.
Response 2. We thank the reviewer for pointing out these relevant papers. We have now added comparison of these results in the discussion section in rows 295-301. We have also compared our imaging results with previous studies in rows 285-287.
- Clearly write the conclusion of the abstract and text. And the content of the abstract should correspond to the content of the text.
Response 3. We thank the reviewer for this important opinion. We agree that the abstract and conclusions did not adequately match and we believe that after the revision the conclusions are now clear and easier to find for the reader. (Rows 10-23, 344-348)
- The table should be in the form of a three-line table. And the content of Table 3 is not very clear.
Response 4. We definitely agree with the reviewer. All the tables are now restructured as three-line tables. CECDAI-row has been moved from Table 3 to text in results in rows 240-241 so that the table now only presents laboratory findings. We hope that the tables are now easier to read and there are fewer empty cells as well.
- Discussion section is too scattered and needs to be more focused
Response 5. We thank the reviewer for this important remark. The discussion section is now also revised and largely rewritten. We have moved many numeric results to the results section for easier reading. We feel that the discussion section is now less scattered and more focused as the chapters are now in more logical order. Our conclusions are now more clearly stated in discussion.